# Multimodal Physical Exercise Affects Visuo-Spatial Working Memory: Preliminary Evidence from a Descriptive Study on Tai-Chi Practitioners and Runners

**DOI:** 10.3390/brainsci13101400

**Published:** 2023-09-30

**Authors:** Emahnuel Troisi Lopez, Marianna Liparoti, Noemi Passarello, Fabio Lucidi, Laura Mandolesi

**Affiliations:** 1Institute of Applied Sciences and Intelligent Systems, National Research Council, 80078 Pozzuoli, Italy; e.troisilopez@gmail.com; 2Department of Social and Developmental Psychology, Faculty of Medicine and Psychology, University of Roma “Sapienza”, 00185 Rome, Italy; marianna.liparoti@gmail.com (M.L.); fabio.lucidi@uniroma1.it (F.L.); 3Department of Humanities, University of Naples Federico II, 80138 Naples, Italy; noemi.passarello@unina.it

**Keywords:** cognitive functions, physical activity, executive functions, sport, memory

## Abstract

Recent evidence has shown a relationship between physical exercise (PE) and cognitive functioning. However, it is unknown if unimodal and multimodal modalities of PE affect cognitive abilities in different ways. To fill this gap, we analyzed the effects of unimodal PE (running) and multimodal PE (Tai Chi) on specific cognitive abilities. A sample of 33 participants (mean age = 52.6 ± 7.2) divided into eleven runners, eleven Tai Chi practitioners, and eleven age-matched sedentary individuals were subjected to a neuropsychological tests battery to assess shifting and problem solving abilities (Rule Shift Cards, BADS-RS, and Key Search tasks), verbal fluency (semantic and phonemic verbal fluency tasks), verbal memory (Rey’s 15 words test), visuo-spatial working memory (Corsi test), and global cognitive functioning (clock-drawing test). The results showed significantly higher BADS-RS scores in runners and Tai Chi practitioners in comparison to the sedentary participants, thus evidencing improved shifting abilities for active individuals. Interestingly, post hoc analysis showed significantly higher span scores of Corsi test only in Tai Chi practitioners as compared to sedentary participants, suggesting how multimodal PE facilitates the visuo-spatial working memory processes. Although preliminary, our descriptive study indicates that the type of PE could modulate specific cognitive domains, even if the practice of motor activity favors a global cognitive improvement.

## 1. Introduction

An active lifestyle based on physical exercise (PE) increases health potential. PE is defined as “a subcategory of physical activity that is planned, structured, repetitive, and aimed at improving or maintaining physical fitness”, determining psychophysical well-being. For this reason, its potential is recognized worldwide [1]. Several scientific studies have evidenced the role of PE in maintaining or improving health status across the lifespan [2,3,4]. In fact, the lack of PE is correlated to neurological, musculoskeletal, and psychological diseases [5,6], while regular PE reduces the risk of dementia, revealing thus an important neuroprotective factor [7]. Moreover, it has been observed how PE is able to act directly on the brain inducing specific neuroplastic phenomena [8,9,10,11]. 

Recently, evidence has been growing that highlights the positive effects of regular PE on cognitive functioning at all ages. 

In children and adolescents, regular PE plays a pivotal role in increasing learning and memory abilities as well as the functionality of attentional processes. For example, Voss et al. [12] observed that young people who practice aerobic exercise obtain better scores in verbal, perceptual, and arithmetic tests than sedentary peers and also have a better academic performance. More recently, Serra et al. [13] showed that young gymnasts performed better than the control group in a spatial memory task. There is multiple evidence of improved cognition in adults who regularly practice PE, both at an amateur and competitive level [8]. Indeed, especially in sportive individuals, better cognitive abilities, such as cognitive flexibility, memory, and attention, are well documented [14,15,16]. Regular PE impacts on cognitive functioning even in the elderly [17]. In fact, active elderly people have more developed planning abilities and executive functions as well as better spatial orientation ability in comparison to elderly sedentary people. 

Currently, ongoing research is attempting to understand the type of PE needed to produce these beneficial effects. The studies on aerobic vs. anaerobic exercise as well as those on acute vs. chronic exercise [18,19], have demonstrated that mainly steady aerobic PE, practiced regularly (chronic exercise), is the one that induces the greatest benefit on the brain and cognitive functioning [8]. However, some positive effects on cognition are emerging also from anaerobic and acute exercise (based on a single bout of exercise) [20,21,22]. In addition to this scientific discussion, the impact of unimodal and multimodal PE on cognitive functions is starting to be debated [23]. Unimodal PE is characterized by a traditional modality of motor exercises, which involves mainly aerobic exercises and repetitive motor gestures. In addition, it is structured to improve muscle strength and endurance without an involvement focused on cognitive processes such as planning, motor imagery, decision making and executive abilities, except for particular sports [23,24,25]. In this line, running, swimming, and cycling, could be considered unimodal PE examples. Differently, multimodal PE is a body–mind intervention that establishes a balance between body and mind and emphasizes the conscious control of every movement and perception of the body in order to improve physical and psychological well-being [23]. For these characteristics, multimodal PEs are considered to be those activities that force cognitive and motor processes simultaneously, such as motor imagery, attention, perception, and motor control. Among multimodal PEs, Tai Chi practice can be considered a body–mind activity. Namely, it is focused on improving motor function (balance, stability, flexibility of movements), mental health (reducing stress, anxiety, and depression), and cognitive functioning [26,27,28,29]. 

Although unimodal PE does not require a great deal of cognitive effort, many studies have documented activities based on aerobic endurance exercise, such as running or walking at sustained speed with moderate intensity, to determine an improvement in cognitive functioning correlated to neuroplasticity [30,31,32,33,34]. In the same way, there is also substantial evidence on the positive effects of multimodal activities, such as Tai Chi and meditation, on cognitive functioning and the brain [35,36].

In this context, it can be interesting to analyze whether unimodal and multimodal activities influence the different cognitive domains in a specific way. To clarify this topic, in the present study, we investigated the influence of unimodal and multimodal PE on specific cognitive abilities, including executive functions, verbal fluency, memory, and visuospatial abilities. In particular, we recruited eleven runners (unimodal PE), eleven Tai Chi practitioners (multimodal PE) and eleven age-matched sedentary individuals.

## 2. Materials and Methods

### 2.1. Participants

We recruited a total of 33 participants (mean age = 52.6 ± 7.2) divided into eleven runners (unimodal PE), eleven Tai Chi practitioners (multimodal PE), and eleven age-matched sedentary participants that had never practiced sport in a structured and constant manner over time, to carry out a descriptive study. For the recruitment, we sent a request for voluntary participation to different local sports associations, universities, and recreative centers. All practitioners had at least 8 continuous years of experience in their own sport and did not practice any other physical activity (Table 1). Furthermore, the weekly hours of activity were similar between Tai Chi practitioners and runners (approximately six and seven weekly hours, respectively). All participants were native Italian speakers.

Inclusion criteria were normal or corrected-to-normal vision, right-handedness, and aged > 18 years. Moreover, participants were considered eligible if they had a score of ≥26 in the Mini-Mental State Examination [37]. Alternatively, exclusion criteria comprised the current or past presence of psychopathology, psychiatric, neurological, or motor disorders, any anxiety symptoms as indicated by the Hamilton Anxiety Scale [38] scores and any history of drug or alcohol abuse, or other medical illness.

The participants were voluntarily enrolled after written informed consent was obtained. The study was approved by the Local Ethical Committee (University of Naples Federico II, n. 26/2020) and was carried out in accordance with the Declaration of Helsinki.

### 2.2. Psychological Assessment

The cognitive assessments were carried out with the aid of several neuropsychological tasks. In particular, we administered two subtests of the Behavioral Assessment of the Dysexecutive Syndrome (BADS) [39]: the Rule Shift Cards (BADS-RS) and the Key Search (BADS-KS). Specifically, BADS-RS uses playing cards in red and black with different rules for different parts of the task. This task allows the assessment of the ability to shift to a new rule and to ignore a previous rule. Scoring is based on the time taken and the number of errors. BADS-KS assesses the ability to plan a strategy to solve a problem (finding a key, lost in a field). The score is based on a number of criteria, such as a systematic or efficient strategy. Additionally, for all participants we examined semantic and phonemic verbal fluency [40,41], aimed at giving a measure of quick word search abilities and the ability to access the semantic-lexical warehouse. The score is based on the number of words produced that begin with a specific letter of the alphabet (verbal fluency) or that can be inserted into a semantic category (semantic fluency). Furthermore, Rey’s 15 words test [42] was used to estimate learning and verbal long-term memory of new information. The score is based on the number of words learned. The clock-drawing test [43,44] was used to assess global cognitive functioning. Namely, to successfully draw a clock, it is necessary to understand verbal instructions, encode instructions in short-term memory, and use visual constructive skills to draw a clock. Finally, the Corsi test [45] was used to evaluate the visuospatial working memory. In this task, the experimenter shows nine blocks arranged in front of the participant and he/she taps a sequence of blocks with different lengths. The participant has to repeat the blocks sequence that the experimenter showed, in the same order. The longest sequence a participant can correctly repeat is the span.

The scores obtained were corrected for age and education. All participants were tested individually in a soundproof room.

### 2.3. Statistical Analysis

Statistical analysis was performed using MATLAB (Mathworks^®^, version R2013a). In order to compare the psychological scores of runners, Tai Chi practitioners, and sedentary individuals, the analysis of variance (ANOVA) was performed followed by post hoc analysis between groups using a two-sample t-test. All the *p*-values were corrected for multiple comparisons using the false discovery rate (FDR) [46]. The statistical significance was defined as *p* < 0.05.

## 3. Results

Demographic, anthropometric, and inclusion/exclusion criteria data that we collected were compared among groups. As showed in Table 2, there were no significant differences among Tai Chi practitioners, runners, and sedentary individuals.

The ANOVA analysis showed a significant difference of the BADS-RS subtest (F (2,30) = 6.333, *p* = 0.005, *p*_FDR_ = 0.030) and Corsi test (F (2,30) = 6.442, *p* = 0.005, *p*_FDR_ = 0.030) among groups.

In detail, the post hoc analysis showed significantly higher scores of BADS-RS subtests in runners (*p* = 0.022) and Tai Chi practitioners (*p* = 0.007) with respect to the sedentary individuals (Figure 1A).

Furthermore, regarding the Corsi test, the post hoc analysis showed significantly higher span in Tai Chi practitioners (*p* = 0.001) as compared to sedentary individuals (Figure 1 B). A tendency towards a significant difference was also highlighted between runner and Tai Chi groups (*p* = 0.057). Finally, other neuropsychological tests failed to produce significant differences among groups: BADS-KS subtest (F (2,30) = 2.09, *p* = 0.141, *p*_FDR_ = 0.423); semantic verbal fluency (F (2,30) = 0.483, *p* = 0.621, *p*_FDR_ = 0.746); phonetic verbal fluency (F (2,30) = 2.97, *p* = 0.066, *p*_FDR_ = 0.266); Rey’s 15 words immediate memory (F (2,30) = 0.268, *p* = 0.766, *p*_FDR_ = 0.805); Rey’s 15 words delayed recall (F (2,30) = 0.218, *p* = 0.805, *p*_FDR_ = 0.805); clock-drawing test (F (2,30) = 0.760, *p* = 0.476, *p*_FDR_ = 0.745).

## 4. Discussion

The present study aimed to examine the effects of unimodal and multimodal PE on cognitive functions, and to investigate which type of physical exercise mostly affects cognitive abilities. By means of specific neuropsychological tasks, we evaluated the cognitive performances of people practicing Tai Chi and running, representing, respectively, multimodal PE (Tai Chi) and unimodal PE (running). Then, we compared their scores with the scores of people who do not practice any sports. In particular, we investigated several cognitive domains, including executive functions, verbal fluency, memory, and visuospatial abilities.

The first finding concerns the evidence that both modalities of PE, unimodal and multimodal activities, positively affect specific facets of executive functions. In particular, Tai Chi practitioners and runners showed a better performance with respect to sedentary individuals in the Rule Shift Cards test. This task is related to the ability to switch between rules when viewing a deck of playing cards, and it requires a great cognitive flexibility, which may be interpreted as the ability to adapt information processing strategies, to cope with new or unexpected conditions and to inhibit (incorrect) responses [39]. Therefore, our data evidenced that PE—in general and not specific modalities—positively affects the shifting abilities. Although our finding should be interpreted with caution due to the sample size, it is supported by scientific evidence in which the effects of running and Tai Chi on cognition were independently examined [47,48,49,50,51]. The improvement of cognitive flexibility in runners and Tai Chi practitioners could depend on the fact that the shifting ability is an experience-dependent process [8]. In fact, in each of these activities, even if in a different way, this cognitive ability is exercised. Specifically, during the running, it is necessary to process continuously the surrounding environment, which, in the case of outdoor activity, changes continuously, thus forcing the runner to make adjustments on his/her route. Therefore, runners need to focus carefully on the new environmental conditions and quickly adapt the information on processing strategy to the variation of stimuli to achieve better physical performance. Our observation is supported by several studies in which it was shown that, during endurance PE, cognitive demands are high because athletes, instead of their mind wandering, focus their attention on the execution of the performance and on the surrounding environment [50,51]. Moreover, it is important to underline that, although unimodal PE, such as running, does not require much cognitive effort, it has been shown that this activity cannot be considered a simple automatic and repetitive activity disconnected from cognition [33]. In fact, running and other unimodal PE, such as walking, involve different cognitive domains including attention, inhibition, planning, and motor control [34]. Tai Chi is also characterized by putting into action exercise of cognitive flexibility. This type of multimodal activity, as well as meditation, includes meditative components determining mental exercise of attention and favouring cognitive flexibility development [36]. In support of this, it is important to note that breath and mind are considered the main components of Tai Chi activity deep breathing being the basis of relaxation. Moreover, Tai Chi activity is considered as a “moving meditation” [52] because, like meditation, it focuses on specific parts of the body, focusing attention from one moment to the next [53]. This continuous task sharpens the attentive processes and determines strong mind control. These meditative characteristics of Tai Chi should allow for awareness training that could positively modulate the capacity to respond to stimuli in a non-automated manner. Indeed, it has been demonstrated that all meditative practices are related to an improvement of attentional processes as well as to cognitive flexibility [54].

Although few studies have directly analysed the effects of running on cognitive flexibility [55], there is evidence documenting regular aerobic exercise affecting cognitive flexibility [32]. Recently, Król and Gruszka [56] observed that orienteering runners had a higher level of cognitive flexibility than other runners, thus suggesting the importance of changes of behavioural strategies in response to changes in the situation in improving this cognitive ability.

For that which concerns the improvements in cognitive flexibility in the Tai Chi group in comparison to sedentary individuals, this finding is consistent with previous studies, in which it was observed that Tai Chi practice affects task-switching abilities [35]. In this context, Lei Cui et al. [35] observed that Tai Chi practice alters the functionality of different brain-network areas with respect to the sedentary group, suggesting that Tai Chi could promote a brain functional specialization which is a predictor of better cognitive flexibility. Behavioral and psychological evidence is supported by long-lasting effects on brain networks in meditators [57,58]. Additionally, it may be considered that Tai Chi is a multimodal exercise usually performed as a group activity therefore it offers the opportunity for social interaction that could also affect the brain and cognition. Indeed, Mortimer et al. [59] observed that Tai Chi practitioners as well as individuals who had received exposure to social factors increased cognitive performances in comparison to a walking group and naïve control group.

Interestingly, moving to our second set of results, we observed an increase of spatial span (Corsi test) only in Tai Chi practitioners, suggesting thus a better visuo-spatial working memory in these individuals than found in those of the other groups. A possible explanation is that Tai Chi practitioners, during their activity, need to recall a long sequence of movement and forms (up to 108 forms), and have to maintain these items in their memory load [60], thus exercising their working memory abilities. Our finding is supported by various studies in which it was demonstrated that exercises performed during Tai Chi, involving choreographic movements, are related to a significant improvement of spatial working memory abilities [61]. In this line, recently Wang et al. [48] showed that Tai Chi practice improves individuals’ working memory capacity.

Although only preliminary, our results evidence that Tai Chi practice has a positive influence on cognitive functioning in general and on visuo-spatial working memory in particular, suggesting thus that this practice proves to be an effective behavioral strategy to improve cognitive abilities. Thus, it could have important implications for clinical and rehabilitative practice, underlining the need to promote alternative and adapted modalities of physical exercise as a strategy to maintain or strengthen different aspects of memory, such as in the case of Alzheimer’s or other neurodegenerative diseases. Furthermore, Tai Chi practice could also be promoted as an intervention strategy to achieve active aging.

Several limitations are present in this study. First, since only one type of unimodal and multimodal activity was compared, it might also be interesting to compare cognitive abilities in different unimodal and multimodal PEs, such as walking, yoga, Pilates, and so on. Second, Tai Chi practice has many different styles and features and these should be considered in future studies. Third, increasing the sample size could offer the opportunity to highlight the differences between the PE modalities and the sedentary group with stronger statistical power. Indeed, due to the COVID-19 pandemic, we had to stop the recordings, and could not reach the sample size that was originally planned.

In conclusion, our study highlights that PE has positive effects on cognitive functioning and, in particular, that Tai Chi, as a multimodal activity, specifically affects visuo-spatial working memory abilities. Research should continue to explore this new line of investigation, as this could aid in the development of physical exercise programs that could be applied to memory disorders and healthy aging.

## Figures and Tables

**Figure 1 brainsci-13-01400-f001:**
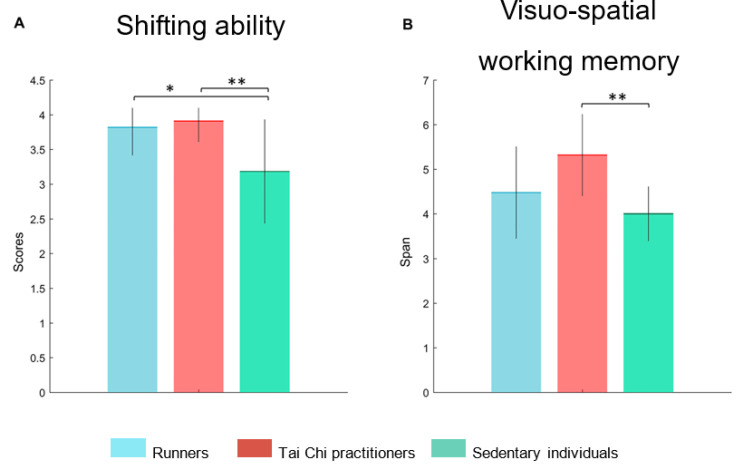
Psychological comparison. (**A**) The histogram refers to the comparison among runners, Tai Chi practitioners, and sedentary individuals, in relation to shifting abilities assessed by the Rule Shift Cards (BADS-RS). The graph shows significantly higher scores of BADS-RS in runners (*p* = 0.022) and Tai Chi practitioners (*p* = 0.007) with respect to sedentary people. (**B**) The histogram refers to the comparison among runners, Tai Chi practitioners, and sedentary individuals in spatial span assessed by Corsi test. The graph shows a significantly higher span in the Tai Chi group (*p* = 0.001) with respect to the sedentary group. Significant *p*-value: * *p* < 0.05, ** *p* < 0.01.

**Table 1 brainsci-13-01400-t001:** Years of practice of Tai Chi and runners relative to participants of both active groups.

Tai Chi Practitioners Years of Practice	Runners Years of Practice
P 1 = 8	P 1 = 9
P 2 = 8	P 2 = 9
P 3 = 11	P 3 = 8
P 4 = 9	P 4 = 8
P 5 = 8	P 5 = 9
P 6 = 9	P 6 = 10
P 7 = 8	P 7 = 11
P 8 = 8	P 8 = 8
P 9 = 8	P 9 = 8
P 10 = 10	P 10 = 10
P 11 = 9	P 11 = 13
Mean (± std dev) = 8.72 (±1.0)	Mean (± std dev) = 9.36 (±1.6)

**Table 2 brainsci-13-01400-t002:** Demographic, anthropometric, and inclusion/exclusion criteria of the participants. Note: Data are given as mean ± standard deviation. Abbreviations: BMI, body mass index; MMSE, mini mental state examination.

Features	Tai Chi	Runner	Sedentary	*p*-Value
Age (years)	56 ± 6.6	49 ± 6.9	52.8 ± 8.1	n.s
Gender (m/f)	5/6	10/1	6/5	-
Education (years)	15.3 ± 2.6	15.2 ± 3.3	14.8 ± 4	n.s
BMI (Kg/m^2^)	25.7 ± 3.7	23.4 ± 1.8	25.1 ± 2.2	n.s
Years of physical practice	9.36 ± 1.5	9.36 ± 1.6		n.s
MMSE	27.9 ± 1	27.5 ± 1.1	27.1 ± 1.4	n.s
Hamilton	5.3 ± 3.6	4.5 ± 5.1	7.5 ± 5.1	n.s

## Data Availability

Data are available under request to the corresponding author: Laura Mandolesi (laura.mandolesi@unina.it).

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
