# Peer review of "Multimodal Physical Exercise Affects Visuo-Spatial Working Memory: Preliminary Evidence from a Descriptive Study on Tai-Chi Practitioners and Runners"

_brainsci, 2023, doi:10.3390/brainsci13101400_

Round 1

Reviewer 1 Report

I liked this work.  However, I did detect grammatical errors when reading over the manuscript.

There were some grammatical errors in the manuscript.  The authors should thoroughly proofread the manuscript.

Author Response

I liked this work.  However, I did detect grammatical errors when reading over the manuscript.

There were some grammatical errors in the manuscript.  The authors should thoroughly proofread the manuscript.

We thank the Reviewer for his/her comment. We checked the text and corrected the grammatical errors.

Reviewer 2 Report

This study investigated the impacts of unimodal and multimodal physical exercise on cognitive functioning and found that multimodal exercise appeared to influence working memory. However, several major issues need to be addressed further.

-       Please provide the information for approval from the Institutional Review Board.

-       How was the sample size (n=11) determined?

-       How many exact years of exercise have been practiced by each participant in different groups?

-       Table 1 should be illustrated in the Result section.

-       How were the psychological assessments conducted? The procedures should be provided in detail.

-       What was the method of post-hoc analysis used? What was the rate of FDR?

-       Line 137, it is not clear how the statistical difference was identified between the runner and tai-chi groups.

-       Line 138, data on negative results should also be reported.

-       The results did not suffice to back up the conclusion.

The language needs to be improved.  

Author Response

This study investigated the impacts of unimodal and multimodal physical exercise on cognitive functioning and found that multimodal exercise appeared to influence working memory. However, several major issues need to be addressed further.

-       Please provide the information for approval from the Institutional Review Board.

We inserted the missing information.

-       How was the sample size (n=11) determined?

Unfortunately, the sample size of our study is limited due to the occurrence of the COVID-19 pandemic, which occurred during the time when we were collecting data. We will include this information among the study's limitations.

-       How many exact years of exercise have been practiced by each participant in different groups?

Here are the specific years of practice. This information has been included in Table 1 as mean +/- standard deviation for each group.

-       Table 1 should be illustrated in the Result section.

We thank the Reviewer 1 for this suggestion. We moved the table 1 (changed in Table 2) to the results section.

-       How were the psychological assessments conducted? The procedures should be provided in detail.

We thank the Reviewer 1. We added missing information about the psychological assessments in the text.

-       What was the method of post-hoc analysis used? What was the rate of FDR?

For the post hoc analysis, we used a two-sample t-test. We didn’t find false positive, as can now be observed with all the comparison results added. All the information has been included in the manuscript.

-       Line 137, it is not clear how the statistical difference was identified between the runner and tai-chi groups.

We apologize, the information was wrongly reported. In that line, we meant to refer to a statistical trend toward significance when comparing the tai chi and the runner groups in the Corsi test. We corrected the information and reported the p value (p = 0.057).

-       Line 138, data on negative results should also be reported.

Results section now display all the statistical results, including negative results.

-       The results did not suffice to back up the conclusion.

We apologize. We modified the discussion and conclusion.

The language needs to be improved.

We hope to have improved the language.

Reviewer 3 Report

Congratulations to the authors for their work.

I think he should change the name of the control group to sedentary, as there is no experimental intervention in this study, but only a descriptive comparison of the practice of physical exercise.

Although this is an interesting study, I believe that the sample is too small to be able to analyse the results in depth.

The methodology should indicate the type of study. Descriptive, how the sample was selected....

It would be interesting to indicate in the title that this is a descriptive study, and it is not clear why only Tai Chi practitioners are mentioned in the title and not runners.

In the abstract it is not clear what type of study. The conclusions should be improved to make them clearer.

Lines 70-72 should be clarified to a greater extent because these activities are considered unimodal with some citation. For example, it is known that aerobic activities have great benefits for the brain.

Why are only two Behavioural Assessment of the Dysex- eecutive Syndrome (BADS) tests administered? If only two subtests are used, can they be considered to assess executive functions as a whole? Would it not be more convenient to indicate the results of each subtest indicating the types of domains of executive functions to which it refers?

Were the participants checked to see whether, in addition to the physical activities in which they were grouped, they practised any other modalities that might interfere with the results? The average weekly time of practice was considered, for example it is not the same to practice the activity twice a week than 6 days a week. This would have been a covariate.

In the discussion a large number of quotations are used that have not been taken into account in the introduction.

The conclusions are not very clear.

Author Response

Congratulations to the authors for their work.

I think he should change the name of the control group to sedentary, as there is no experimental intervention in this study, but only a descriptive comparison of the practice of physical exercise.

Although this is an interesting study, I believe that the sample is too small to be able to analyse the results in depth.

The methodology should indicate the type of study. Descriptive, how the sample was selected....

We thank the Reviewer 3. We added this information in the manuscript (“2.1 Participants” section). In particular, we have sent a request for voluntary participation to different local sports associations.

It would be interesting to indicate in the title that this is a descriptive study, and it is not clear why only Tai Chi practitioners are mentioned in the title and not runners.

We thank the Reviewer 3. In according to his/her observation, we suggested another title.

In the abstract it is not clear what type of study. The conclusions should be improved to make them clearer.

We thank the Reviewer 3. In the abstract, we specified the type of the study, and we hope to have improved the conclusion.

Lines 70-72 should be clarified to a greater extent because these activities are considered unimodal with some citation. For example, it is known that aerobic activities have great benefits for the brain.

We thank the Reviewer for giving us the opportunity to fully clarify the conceptual difference between unimodal and multimodal exercises. To support this, as suggested by the Reviewer, we included the following references in the manuscript:

Govindaraj, R., Karmani, S., Varambally, S., & Gangadhar, B. N. (2016). Yoga and physical exercise – a review and comparison. International Review of Psychiatry, 28(3), 242–253. https://doi.org/10.3109/09540261.2016.1160878

Liparoti, M. (2021). Effects of acute and chronic, multimodal and unimodal, physical exercise on brain of elderly people: a systematic review. Giornale Italiano di Educazione alla Salute, Sport e Didattica Inclusiva, 5(2).

Morat, M., Morat, T., Zijlstra, W., & Donath, L. (2021). Effects of multimodal agility-like exercise training compared to inactive controls and alternative training on physical performance in older adults: A systematic review and meta-analysis. European Review of Aging and Physical Activity, 18, 4. https://doi.org/10.1186/s11556-021-00256-y

Why are only two Behavioural Assessment of the Dysex- eecutive Syndrome (BADS) tests administered? If only two subtests are used, can they be considered to assess executive functions as a whole? Would it not be more convenient to indicate the results of each subtest indicating the types of domains of executive functions to which it refers?

We thank the Reviewer for his/her comment. We are in perfect agreement and have modified the text specifying the type of cognitive ability assessed.

Were the participants checked to see whether, in addition to the physical activities in which they were grouped, they practised any other modalities that might interfere with the results? The average weekly time of practice was considered, for example it is not the same to practice the activity twice a week than 6 days a week. This would have been a covariate.

We ensured that all participants were not engaged in other specific physical activities that could influence the results. Additionally, the hours dedicated to practice were very similar among them. The runners trained for approximately 7 hours per week, while the tai chi practitioners trained for about 6 hours per week. These information have been reported into the manuscript (“2.1 Participants” section).

In the discussion a large number of quotations are used that have not been taken into account in the introduction.

We modified the text in according to Reviewer observation.

The conclusions are not very clear.

We changed the conclusion in order to clarify our conclusions.

Round 2

Reviewer 2 Report

The authors made efforts to improve this manuscript. One point still needs to be clarified. Line 209-210, the authors stated that due to the pandemic, the sample size could not be increased, which is inappropriate. The sample size should be predetermined before the start of enrollment of participants. However, it is possible that the number of participants in this study was limited by the pandemic. Please revise.

Author Response

The authors made efforts to improve this manuscript. One point still needs to be clarified. Line 209-210, the authors stated that due to the pandemic, the sample size could not be increased, which is inappropriate. The sample size should be predetermined before the start of enrollment of participants. However, it is possible that the number of participants in this study was limited by the pandemic. Please revise.

We thank the Reviewer for his/her comment. We revised this point. Namely, the pandemic has limited us in continuing the programmed enrollment. We rephrased the lines".

Reviewer 3 Report

Congratulations to the authors for their work in improving the article.

Author Response

Congratulations to the authors for their work in improving the article.

We thank the Reviewer for his/her appreciation.